# Demographic Characteristics and Economic Burden of *Clostridioides difficile* Infection in Korea: A Nationwide Population-Based Study after Propensity Score Matching

**DOI:** 10.3390/antibiotics13060542

**Published:** 2024-06-10

**Authors:** Jae Myung Cha, Jin Young Yoon, Min Seob Kwak, Moonhyung Lee, Young-Seok Cho

**Affiliations:** 1Department of Internal Medicine, Kyung Hee University Hospital at Gangdong, Kyung Hee University College of Medicine, Seoul 05278, Republic of Korea; htherehthere@khu.ac.kr (J.Y.Y.); kwac63@khu.ac.kr (M.S.K.); 02230038@khnmc.or.kr (M.L.); 2Department of Internal Medicine, Seoul St. Mary’s Hospital, College of Medicine, The Catholic University of Korea, Seoul 06591, Republic of Korea

**Keywords:** *Clostridioides difficile*, demography, hospital costs, longitudinal studies, propensity score

## Abstract

*Clostridioides difficile* infection (CDI) poses a considerable threat to global public health. However, there have been insufficient propensity score-matched data on its demographic characteristics and economic burden. Using nationwide claims data, we assessed longitudinal changes in the demographic characteristics and economic burden of CDI between 2011 and 2019 after propensity score matching. We performed a regression analysis to compare the differences in the length of hospital stay and medical costs between patients with CDI and controls (gastroenteritis and colitis). The CDI hospitalization rate increased 2.9-fold between 2011 and 2019. The CDI group had higher comorbidity index scores and was more frequently diagnosed at tertiary hospitals and in the Seoul region than the control group (all *p* < 0.001). The annual incidence rate of CDI/10,000 persons significantly increased in both sexes and all age groups. The length of hospital stay and medical costs were 3.3-fold and 5.0-fold greater, respectively, in the CDI than in the control group (both *p* < 0.001). Although the length of hospital stay decreased, total medical costs increased in all age groups and both sexes between 2011 and 2019 (all *p* < 0.001). When compared with the control group, the CDI-attributable length of hospital stay and medical cost were greater by 15.3 days and KRW 3413 (×10^3^), respectively, after matching. In conclusion, CDI incidence, particularly among the elderly population with comorbidities, has been increasing. In addition, the length of hospital stay and total medical costs of the CDI group were greater than those of the control group.

## 1. Introduction

*Clostridioides difficile* (CD) is a spore-forming anaerobic Gram-positive bacillus that is widely distributed in nature [1]. CD infection (CDI) can cause variable clinical symptoms ranging from mild diarrhea to fatal colitis [1], and it is known as the most common cause of hospital-acquired diarrhea, especially in antibiotics users [2]. Furthermore, CDI can lead to treatment failure, increased medical costs, long-term hospitalization, and increased socioeconomic burden [2]. Because CDI is a potentially preventable disease, effective prevention, rapid and accurate diagnosis, and appropriate use of antibiotics are necessary to reduce its spread in medical institutions.

CDI has been globally recognized as a considerable threat to public health, with its incidence increasing worldwide [3,4,5,6]. CDI increased by 4.4-fold in a region of Quebec, Canada, from 1991 to 2003 [3], and by 2.1-fold in a multi-institutional outbreak in the USA from 1993 to 2003 [4]. However, domestic epidemiological data are necessary, as CDI incidence may vary from country to country because of varying antibiotic use patterns. In Korea, two nationwide datasets reported its incidence based on the Health Insurance Review and Assessment (HIRA) database [5,6]. Choi et al. [5] reported that the annual prevalence of CDI increased from 1.43/100,000 persons in 2008 to 5.06/100,000 in 2011. Kim et al. [6] also reported a significantly increased incidence of CDI between 2008 and 2020, from 0.3 to 1.8 per 10,000 patient days, 0.3 to 1.6 per 1000 admissions, and 6.9 to 56.9 per 100,000 population, respectively. However, previous studies were limited by the lack of propensity score (PS) matching [5,6] or a short study period [5]. There have been insufficient data on CDI for well-designed, PS-matched analysis using representative Korean nationwide data [7]. For the development of effective strategies for CDI, it is essential to understand the current disease states of CDI.

Until now, only a few studies have evaluated the economic burden of CDI. The economic burden of CDI is expected to increase significantly with the increasing burden of main risk factors of CDI, such as old age, antibiotic use, and long-term hospital stays. South Korea is a rapidly aging country in line with global trends. In South Korea, 19.2% of the total population is elderly, older than 65 years, and this proportion is expected to increase to 25.3% by 2030 [8]. Long-term hospital stays are often noticed in older patients due to comorbid diseases, frailty and malnutrition, as well as CDI. In addition, a health insurance database-generated time series analysis reported the temporal change in antibiotic use from 2007 to 2014 in South Korea [9]. Defined daily doses of antibiotic use increased from 33.95 in 2007 to 37.32 in 2014 in elderly populations older than 65 years and 23.5 in 2007 to 27.7 in 2014 in all age groups. As our expectation, a Korean study reported that the medical cost of CDI increased by 6.6-fold from 2008 to 2011 [5]. However, the previous study was limited as they included indirect costs, which vary depending on each patient and are difficult to accurately estimate. Considering the increasing incidence of CDI and associated economic burden, updated nationwide, the economic burden of CDI should be evaluated.

In this study, we evaluated the demographic characteristics and economic burden of CDI using nationwide data after PS-matching between 2011 and 2019.

## 2. Results

Figure 1 shows a flowchart of the patients included from the HIRA database. Of the 2,315,789 patients, 123,847 in the CDI group and 2,123,830 in the control group met the eligibility criteria before PS matching, and 123,847 in the CDI group and 247,694 in the control group were included after 1:2 PS matching.

### 2.1. Baseline Characteristics of the Study Population

Table 1 shows the distribution of baseline characteristics in the CDI and control groups before and after PS matching. Sex, age, age group, and annual hospitalizations were well-balanced between the two groups (all SMD < 0.1). In the CDI group, the mean age was 71.8 years, and there was a slight female predominance (55%). The incidence of CDI consistently increased from those in their 20s to those in their 70s but decreased in those in their 80s. The proportion of CDI in the 70s (31.8%) and 80s (29.4%) groups was 61.2%. The annual hospitalization rate increased by 2.9-fold from 6.0% in 2011 to 17.6% in 2019.

### 2.2. Comorbidity, Hospital, and Regional Distribution of CDI

Table 2 shows the comorbidity, hospital, and regional distribution of patients with CDI compared with the control group. Compared to the control group, the CDI group had higher Charlson Comorbidity Index scores and was more frequently detected at tertiary hospitals (both *p* < 0.001). Appendix A shows the nationwide regional distribution of the CDI incidence. In South Korea, 18.1% of CDI was detected in Seoul, while only 1.3% of CDI was detected in Ulsan city.

### 2.3. Annual Occurrence Rate of CDI According to Sex and Age Groups

Table 3 shows the annual occurrence rates of CDI between 2011 and 2019 according to sex and age. Compared with 2011 as a reference, CDI/10,000 persons increased by 2.6-fold in 2019, and this pattern was also observed in both male and female subgroups (Figure 2A). The annual occurrence of CDI/10,000 persons was significantly higher in those over 80 and 70 than in those in their 20s and 30s during the study period (both *p* < 0.001) (Figure 2B).

### 2.4. Length of Hospital Stay and Total Medical Cost of CDI

The length of hospital stay was 3.3-fold greater in the CDI group than in the control group (31.5 vs. 9.6 days, *p* < 0.001), and the medical cost was also 5.0-fold greater in the CDI group than in the control group (KRW 9495 vs. 1891 × 10^3^, *p* < 0.001). The length of hospital stay decreased, but the total medical costs increased in all age groups between 2011 and 2019 (both *p* < 0.001, Figure 3). In addition, the length of hospital stay significantly decreased, but the total medical costs significantly increased in both sexes between 2011 and 2019 (both *p* < 0.001; Appendix A). The length of hospital stay was significantly longer in the male, older, and more comorbid subgroups (Table 4, all *p* < 0.001). The total medical costs were also significantly higher in the male, older, and more comorbid subgroups (all *p* < 0.001). Regarding the type of hospital, the length of hospital stay was significantly lower in tertiary hospitals, but the total medical costs were significantly higher in tertiary hospitals than in other types of hospitals (both *p* < 0.001, Appendix A). When comparing the length of hospital stay and medical cost according to the comorbidity index between CDI and the control group (Table 5), as comorbidities affect the length of stay and medical cost, they were significantly greater in the CDI group than the control group in all comorbidity scores (all *p* < 0.001). The length of hospital stay was longer by 2.2–3.0-fold and the medical cost was also higher by 4.1–5.4-fold in the CDI group than the control group regardless of comorbidity scores. The length of hospital stay was not significantly affected by comorbidity scores.

### 2.5. Length of Hospital Stay and Total Medical Cost of CDI with Regression Models

Appendix A, Appendix A shows the length of hospital stay and total medical cost of CDI using regression models 1, 2, and 3. Assuming sex, age group, and year of hospitalization with PS matching (Model 3), the CDI group showed an increased length of hospitalization by 15.3 days compared with the control group (*p* < 0.001), which explained 42.1% of the variance. Using the same model (Model 3), the CDI group showed an increase in the total medical costs by 3413 (×10^3^) compared with the control group (*p* < 0.001), which is explained by 63.4% of the variance.

## 3. Discussion

To the best of our knowledge, this is the first study to evaluate the demographic characteristics and economic burden of CDI using PS matching in Korea. Our study is unique for its use of nationwide data with PS matching with controls, which included patients with gastroenteritis and colitis of infectious or unspecified origin. Annual hospitalizations for CDI increased by 2.9-fold from 6.0% in 2011 to 17.6% in 2019, compared with a 2.2-fold increase in the control group during the same period. The CDI group had higher Charlson Comorbidity Index scores and was more frequently detected at tertiary hospitals and in the Seoul region than the control group (all *p* < 0.001). The regional difference in CDI occurrence appears to be due to regional differences in hospital distribution rather than actual difference in incidence rate. In Korea, there is an unbalanced regional distribution of tertiary and general hospitals, where 34.0% and 52.3% of CDI cases were detected: 29.8% of tertiary hospitals and 13.1% of general hospitals are located in Seoul (where 18.1% of CDI was reported), while only 2.1% of tertiary hospitals and 2.5% of general hospitals are located in Ulsan (where only 1.3% of CDI was reported). The annual occurrence rate of CDI/10,000 persons significantly increased in both sexes and all age groups from 2011 to 2019 (Figure 2), and it was significantly higher in individuals over 80 and 70 years than in those in their 20s and 30s during the study period (both *p* < 0.001).

Our findings are consistent with previous reports that CDI has become a considerable threat to global public health [3,4,5,6]. Two nationwide datasets of CDI have been reported in Korea [5,6]. Choi et al. [5] reported that the annual prevalence of CDI increased from 1.43/100,000 persons in 2008 to 5.06/100,000 in 2011, based on the HIRA database. However, it is difficult to directly compare these data with our findings because they reported the prevalence of CDI, including inpatients and outpatients, based on calculations from the National Statistical Office data [5]. In addition, they showed only four-year trends without showing long-term trends and matching with a control group. More recently, Kim et al. [6] reported that the incidence of CDI significantly increased from 2008 to 2020, based on the HIRA database: from 0.3 to 1.8 per 10,000 patient-days, 0.3 to 1.6 per 1,000 admissions, and 6.9 to 56.9 per 100,000 population, respectively, during the study period. However, this study was also limited as the primary endpoint was active surveillance of CDI for 3 months at 7 general hospitals and 12 tertiary hospitals, and HIRA data analysis was briefly presented as reference data without a detailed description of the study methodology [6]. In a Canadian study [10], CDI cases were matched to non-CDI cases (those without a positive CD test or clinical CDI); however, it only focused on the length of hospital stay and attributable medical costs rather than epidemiological data. Considering that there have been no PS-matched analyses of the epidemiology of CDI, this study has many significant implications. Increasing CDI incidence in our study from 2011 to 2019 is consistent with a previous study showing that the incidence of CDI in Asia has significantly increased from 2006 to 2014 [11]. The increase in CDI incidence in our study may be explained by patient and virulence factors. For the patient factor, the mean age of patients admitted to hospital in South Korea increased considerably, as did the proportion with numerous comorbidities during the study period. Furthermore, increases in CDI incidence may be largely attributed to the emergence of a more virulent strain, which had been demonstrated in a previous study [12]. However, our study could not analyze the virulence of CDI, so additional studies are needed.

In our study, the length of hospital stay and total medical costs were 3.3-fold and 5.0-fold greater in patients with CDI than in controls (both *p* < 0.001). When comparing the length of hospital stay and medical costs according to the comorbidity index between the CDI and control groups, as comorbidities may affect hospital stay and medical cost, they were significantly greater in the CDI group than the control group in all comorbidity scores (all *p* < 0.001, Table 5). The length of hospital stay decreased, but total medical costs increased in all age groups and both sexes between 2011 and 2019 (all *p* < 0.001). In addition, the length of hospital stay and total medical costs were significantly greater in the male, older, and more comorbid subgroups (all *p* < 0.001). Regarding the type of hospital, the length of hospital stay was significantly lower in tertiary hospitals, but the total medical costs were significantly higher in tertiary hospitals than in other types of hospitals (both *p* < 0.001). The economic burden of CDI was also reported in the National Health Insurance Service-National Sample Cohort from 2006 to 2015 in Korea [13]. In this PS-matched analysis with hospitalized non-CDI cases, the CDI attributable length of hospital stay was 36.9 days, and the medical cost was USD 8298 (KRW 9957.6 × 10^3^). In our regression model, the CDI-attributable length of hospital stay and medical cost were greater by 15.3 days and KRW 3413 (×10^3^), respectively, after PS matching with hospitalized gastroenteritis and colitis of infectious or unspecified origin. A previous study used non-CDI cases as the control [13], but we used different definitions of control (gastroenteritis and colitis of infectious or unspecified origin). Another Korean study also reported that the medical cost of CDI increased by 6.6-fold from USD 2.4 million in 2008 to USD 15.8 million in 2011 [5]. However, this previous study differed from ours in that it included indirect medical costs. Our findings were similar to that of the previous study [5] in that the medical costs of CDI steadily increased in the elderly population and in men. In a Canadian population-based study, CDI cases were matched with non-CDI cases (those without a positive CD test or clinical CDI), and the total adjusted cost and adjusted length of hospital stay were 27% and 13% greater, respectively, than in non-CDI cases [10].

The use of the HIRA database enabled us to perform a nationwide study with PS matching that assessed the demographic characteristics and economic burden of CDI. The results were virtually free from referral bias and readily generalizable owing to the population-based design. However, this study has some limitations. First, we concede that one of the limitations of our study was the use of secondary data with uncertainty regarding the accuracy of diagnosis. However, we used the diagnosis coding of A047 and the prescription history of metronidazole or vancomycin, which may be more accurate than the previous definition that used only A047 coding [5,6,10]. In addition, because we excluded episodes with medical costs or a length of hospital stay more than three times the SD from the mean value, biased cases resulting from comorbid diseases rather than CDI itself may be excluded in our analysis, which is different from those of previous studies [5,10]. Second, detailed information on the chart review level on CDI management was not addressed because of the lack of such data in the HIRA database. Third, we defined controls as individuals with gastroenteritis and colitis of infectious or unspecified origin, which is different from previous studies defining controls as non-CDI patients [10,13]. However, non-CDI controls may not be appropriate for analyzing CDI cases because CDI cases usually have more comorbidities, and it is natural that CDI cases would have a greater length of hospital stay or medical costs than non-CDI cases. Therefore, we compared other definitions of control, gastroenteritis and colitis of infectious or unspecified origin, and judged our definition of control as a more meaningful control group. However, there may be a need for an agreement on the control group for CDI research. Finally, we did not include outpatients with CDI or indirect costs as total medical costs in the current study. However, these were inaccurate and susceptible to other variables; therefore, they were not included in this study.

## 4. Materials and Methods

### 4.1. Data Source

This study is a retrospective nationwide population-based study using the HIRA database, which contains all inpatient and outpatient nationwide data in South Korea [14]. The HIRA database provides comprehensive healthcare coverage for all Koreans and contains information on claims billed by physicians for services, admissions, diagnoses, procedures, discharge status, and patient demographics. Diseases and procedures in claims-based databases had a very high level of agreement with hospital clinical data in South Korea [15,16]. To prevent data errors from separate claims, the analysis conducted was based on hospitalization episodes rather than individual patients in the current study. Episode construction is the process of integrating successive claims while considering the continuity of the medical treatment process. An episode-based analysis is necessary because it is difficult to obtain all medical information with separate claims because hospitals claim separately on a monthly basis when patients are hospitalized for more than one month.

### 4.2. Study Population

We obtained nationwide claims data for 10 years (2010–2019) from the HIRA database with permission (M20210916515). All hospitalized patients diagnosed with A047, A090, or A099 in the International Classification of Disease diagnostic codes as the main diagnosis and sub-diagnosis were included. The CDI group included hospitalized patients newly diagnosed with A047 (enteritis caused by *C. difficile* or pseudomembranous colitis) and prescribed metronidazole or vancomycin (Figure 1). The control group included hospitalized patients who were newly diagnosed with A090 (gastroenteritis and colitis of infectious origin) or A099 (gastroenteritis and colitis of unspecified origin) and were not diagnosed with A047 or prescribed metronidazole or vancomycin (Figure 1). In both groups, episodes with a length of hospital stay or medical costs more than three times the standard deviation (SD) from the mean value were excluded to avoid bias from comorbidities. Patients hospitalized for 0 days were excluded from the analysis. The CDI and control groups were matched in a 1:2 ratio with the PS group for age, sex, and year of hospitalization. The comorbidity index was excluded from the matching variables because it was biased and affected the performance. For patients who were hospitalized multiple times, the index case was defined as the first hospitalization within a year. Because the information used in this study was related only to pseudonyms, the requirement for informed consent was waived. This study was approved by the Institutional Review Board of Kyung Hee University Hospital in Gangdong, Seoul, Republic of Korea (KHNMC 2021-07-012).

### 4.3. Study Variables

Study variables were extracted by coding at the time of hospitalization and included sex, age, age groups, Charlson Comorbidity Index, year of hospitalization (2011–2019), type of hospital (tertiary hospital, general hospital, or convalescent hospital), and regional distribution of CDI occurrence. Dental and oriental hospitals (traditional medicine hospitals prescribing herbal drugs or practicing acupuncture) were excluded. To extract comorbidity data from the year before the index date, data from 2010 was excluded. CDI was compared with the control groups for comorbidity index, type, and regional distribution of hospitalization after PS matching for sex, age, age groups, and year of hospitalization. Longitudinal time changes in the annual occurrence, length of hospital stay, and total medical costs of CDI were assessed from 2011 to 2019. As for the incidence rate per population, annual persons claimed from the HIRA database were used as the denominator because the HIRA database covers almost the entire population of South Korea [5,14].

### 4.4. Statistical Analysis

Continuous variables were presented as the mean ± SD and compared using two-sample *t*-tests. Categorical variables were presented as numbers (percentages) and were compared using the X^2^ test. For PS matching between the two groups, the ‘MatchIt’ package in the R program was used and an SMD (standard mean difference) less than 0.1 after matching was considered indicative of good balance between the groups. The data from 2019 was compared with that from 2011 as a reference. A two-way analysis of variance (ANOVA) was used to determine changes in the length of hospital stay and medical costs according to the main variables. Regression analysis was also performed to compare the difference in the length of hospital stay and total medical cost between the CDI and control groups using models 1–3. Model 1 does not use control variables; Model 2 uses all variables as control variables without considering matching results; and Model 3 assumes that sex, age group, and year of hospitalization are consistent with matching. All statistical tests were two-sided, and a *p* value < 0.05 was considered statistically significant. All statistical analyses were conducted using the SAS Enterprise Guide software (version 9.4.2; SAS Institute Inc., Chicago, IL, USA) and R software packages (R version 3.5.1, www.r-project.org, accessed on 13 April 2024).

## 5. Conclusions

There has been a recent increase in the incidence of CDI, particularly among the elderly population with comorbid diseases. In addition, the length of hospital stay and total medical costs for CDI were greater than those for infectious or unspecified gastroenteritis and colitis. Therefore, effective interventions should be developed to reduce the occurrence of CDI, which is a preventable disease.

## Figures and Tables

**Figure 1 antibiotics-13-00542-f001:**
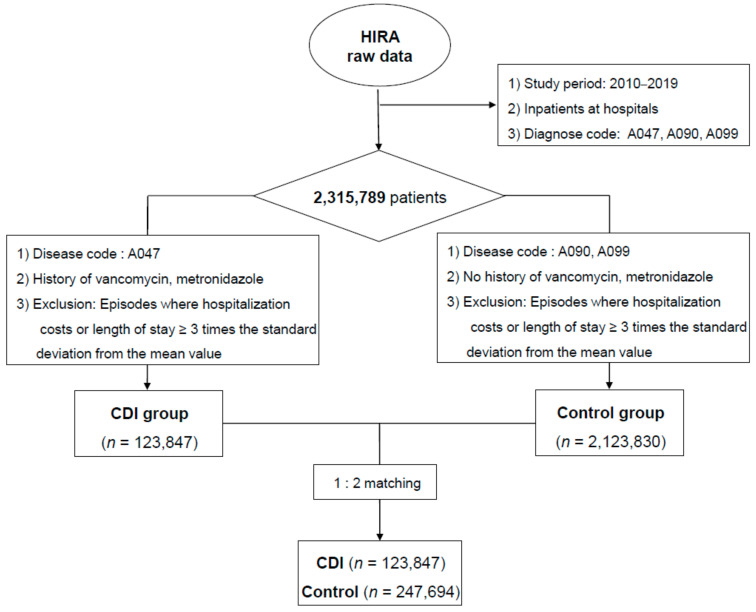
The study flowchart of the included patients. Among a total of 123,847 individuals in the *Clostridioides difficile* infection (CDI) group and 2,123,830 individuals in the control group, 123,847 individuals are included in the CDI group and 247,694 individuals in the control group after 1:2 propensity score matching.

**Figure 2 antibiotics-13-00542-f002:**
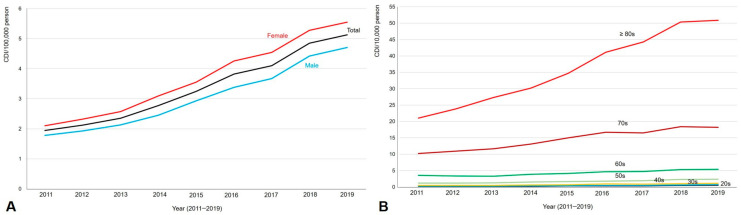
*Clostridioides difficile* infection occurrence between 2011 and 2019 according to sex (**A**) and age groups (**B**). The annual occurrence of CDI is significantly increased in both sexes, and it increased in the over 80 and 70s subgroups compared to the 20s and 30s subgroups (both *p* < 0.001).

**Figure 3 antibiotics-13-00542-f003:**
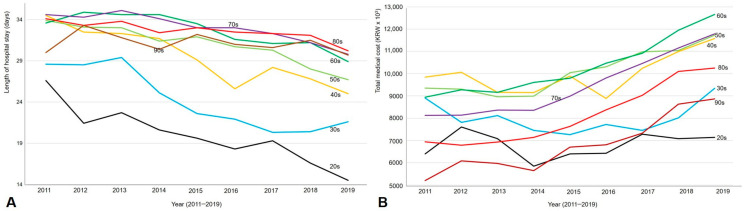
Length of hospital stay (**A**) and total medical cost (**B**) of CDI management between 2011 and 2019. The length of hospital stay is decreased, but the total medical cost is increased in all age groups during the study period.

**Table 1 antibiotics-13-00542-t001:** Baseline characteristics of the CDI and control groups before and after propensity score matching.

Baseline Characteristics	Before Propensity Score Matching			After Propensity Score Matching		
	CDI	Control	SMD	CDI	Control	SMD
Total number of patients	123,847	2,123,830		123,847	247,694	
Sex (male), *n* (%)	55,740 (45.0)	885,136 (41.7)	0.067	55,740 (45.0)	111,480 (45.0)	<0.001
Age, mean (SD)	71.8 (14.4)	53.7 (19.6)	1.050	71.8 (14.4)	71.8 (14.4)	<0.001
Age group (years), *n* (%)			1.051			<0.001
20–29	1961 (1.6)	321,921 (15.2)		1961 (1.6)	3922 (1.6)	
30–39	2900 (2.3)	275,762 (13.0)		2900 (2.3)	5800 (2.3)	
40–49	5531 (4.5)	289,413 (13.6)		5531 (4.5)	11,062 (4.5)	
50–59	12,292 (9.9)	376,467 (17.7)		12,292 (9.9)	24,584 (9.9)	
60–69	19,375 (15.6)	306,482 (14.4)		19,375 (15.6)	38,750 (15.6)	
70–79	39,346 (31.8)	322,361 (15.2)		39,346 (31.8)	78,692 (31.8)	
80–89	36,368 (29.4)	203,063 (9.6)		36,368 (29.4)	72,736 (29.4)	
90–99	6074 (4.9)	28,361 (1.3)		6074 (4.9)	12,148 (4.9)	
Annual hospitalization, *n* (%)			0.190			<0.001
2011	7438 (6.0)	176,149 (8.3)		7438 (6.0)	14,876 (6.0)	
2012	8257 (6.7)	183,266 (8.6)		8257 (6.7)	16,514 (6.7)	
2013	9268 (7.5)	191,782 (9.0)		9268 (7.5)	18,536 (7.5)	
2014	11,079 (9.0)	194,616 (9.2)		11,079 (9.0)	22,158 (9.0)	
2015	13,112 (10.6)	204,352 (9.6)		13,112 (10.6)	26,224 (10.6)	
2016	15,648 (12.6)	307,560 (14.5)		15,648 (12.6)	31,296 (12.6)	
2017	17,000 (13.7)	293,745 (13.8)		17,000 (13.7)	33,997 (13.7)	
2018	20,283 (16.4)	293,560 (13.8)		20,283 (16.4)	40,571 (16.4)	
2019	21,762 (17.6)	278,800 (13.1)		21,762 (17.6)	43,522 (17.6)	

CDI, *Clostridioides difficile* infection; SMD, standardized mean difference; SD, standard deviation.

**Table 2 antibiotics-13-00542-t002:** Comorbidity, hospital, and regional distribution of patients with CDI compared to the control group.

Baseline Characteristics	Before Propensity Score Matching			After Propensity Score Matching		
	CDI	Control	*p* Value	CDI	Control	*p* Value
Total number of patients	123,847	2,123,830		123,847	247,694	
Charlson comorbidity index, *n* (%)			<0.001			<0.001
0	101,891 (82.3)	2,118,162 (99.7)		101,891 (82.3)	247,078 (99.8)	
1	5101 (4.1)	2082 (0.1)		5101 (4.1)	178 (0.1)	
2	5059 (4.1)	1427 (0.1)		5059 (4.1)	168 (0.1)	
≥3	11,796 (9.5)	2159 (0.1)		11,796 (9.5)	270 (0.1)	
Hospital distribution, *n* (%)			<0.001			<0.001
Tertiary hospital	42,074 (34.0)	223,676 (10.5)		42,074 (34.0)	24,637 (10.0)	
General hospital	64,792 (52.3)	1,255,998 (59.1)		64,792 (52.3)	143,041 (57.8)	
Hospital	15,085 (12.2)	618,574 (29.1)		15,085 (12.2)	72,894 (29.4)	
Convalescent hospital	1896 (1.5)	25,582 (1.2)		1896 (1.5)	7122 (2.9)	
Regional distribution			<0.001			<0.001
Seoul	22,404 (18.1)	288,846 (13.6)		22,404 (18.1)	30,289 (12.2)	
Busan	8881 (7.2)	163,223 (5.9)		8881 (7.2)	19,523 (7.9)	
Incheon	4849 (3.9)	126,248 (5.9)		4849 (3.9)	12,736 (5.1)	
Daegu	9960 (8.0)	96,903 (4.6)		9960 (8.0)	12,176 (4.9)	
Gwangju	6535 (5.3)	102,580 (4.8)		6535 (5.3)	10,636 (4.3)	
Daejeon	6473 (5.2)	52,219 (2.5)		6473 (5.2)	5994 (2.4)	
Ulsan	1581 (1.3)	48,308 (2.3)		1581 (1.3)	4777 (1.9)	
Gyeonggi-do	18,849 (15.2)	435,190 (20.5)		18,849 (15.2)	46,208 (18.7)	
Gangwon-do	4100 (3.3)	76,570 (3.6)		4100 (3.3)	10,787 (4.4)	
Chungcheongbuk-do	3565 (2.9)	57,013 (2.7)		3565 (2.9)	7767 (3.1)	
Chungcheongnam-do	3439 (2.8)	92,195 (4.3)		3439 (2.8)	12,061 (4.9)	
Jeollabuk-do	9133 (7.4)	108,057 (5.1)		9133 (7.4)	13,799 (5.6)	
Jeollanam-do	5674 (4.6)	133,918 (6.3)		5674 (4.6)	18,146 (7.3)	
Gyongsangbuk-do	6631 (5.4)	111,377 (5.2)		6631 (5.4)	15,332 (6.2)	
Gyongsangnam-do	9333 (7.5)	194,465 (9.2)		9333 (7.5)	23,101 (9.3)	
Jeju-do	2424 (2.0)	35,757 (1.7)		2424 (2.0)	4250 (1.7)	
Sejong	16 (0.0)	961 (0.1)		16 (0.0)	112 (0.1)	

CDI, Clostridioides difficile infection.

**Table 3 antibiotics-13-00542-t003:** Annual CDI occurrence rates according to sex and age.

	2011	2012	2013	2014	2015	2016	2017	2018	2019
According to sex									
National population									
Total	38,339,076	38,893,731	39,448,916	40,016,161	40,456,410	40,965,509	41,416,899	41,825,145	42,418,657
Male	19,098,543	19,364,589	19,650,074	19,946,756	20,175,946	20,427,764	20,642,446	20,843,532	21,162,592
Female	19,240,533	19,529,142	19,798,842	20,069,405	20,280,464	20,537,745	20,774,453	20,981,613	21,256,065
CDI/10,000 persons									
Total	1.94	2.12	2.35	2.77	3.24	3.82	4.10	4.85	5.13
Male	1.78	1.93	2.13	2.45	2.93	3.38	3.67	4.42	4.71
Female	2.10	2.32	2.57	3.09	3.55	4.26	4.54	5.28	5.55
According to age groups									
National population									
20–29 years	6,772,641	6,688,951	6,664,531	6,729,202	6,781,715	6,845,235	6,892,655	6,905,433	6,936,797
30–39 years	8,265,460	8,193,608	8,073,843	7,906,987	7,785,185	7,674,656	7,534,136	7,454,262	7,331,806
40–49 years	8,613,554	8,639,087	8,752,108	8,796,952	8,732,931	8,696,072	8,628,547	8,441,160	8,407,610
50–59 years	7,273,734	7,559,008	7,808,447	8,023,396	8,134,255	8,249,304	8,327,327	8,473,898	8,590,137
60–69 years	4,014,948	4,141,409	4,294,057	4,532,829	4,879,788	5,185,042	5,471,136	5,763,020	6,136,528
70–79 years	2,526,898	2,727,211	2,833,556	2,909,707	2,926,917	2,993,509	3,133,874	3,251,119	3,348,652
≥80 years	871,841	944,457	1,022,374	1,117,088	1,215,619	1,321,691	1,429,224	1,536,253	1,667,127
CDI/10,000 persons									
20–29 years	0.22	0.19	0.20	0.23	0.32	0.36	0.35	0.46	0.54
30–39 years	0.25	0.25	0.26	0.35	0.43	0.48	0.49	0.57	0.70
40–49 years	0.46	0.45	0.49	0.57	0.68	0.83	0.84	0.97	1.12
50–59 years	1.16	1.19	1.24	1.50	1.65	1.77	1.92	2.29	2.38
60–69 years	3.56	3.35	3.31	3.88	4.14	4.66	4.69	5.31	5.40
70–79 years	10.23	10.97	11.69	13.12	14.98	16.72	16.51	18.38	18.21
≥80 years	21.01	23.84	27.31	30.19	34.71	41.14	44.24	50.34	50.86

**Table 4 antibiotics-13-00542-t004:** Duration of hospital stay and total medical costs associated with CDI.

		Mean (SD)	*p* Value	Quantile (25%–75%)
Length of hospital stay (days)				
Sex	Male	33.1 (27.4)	<0.001	14.0–43.0
	Female	30.2 (25.9)		13.0–39.0
Age groups (years)	20–29	18.9 (21.9)	<0.001	6.0–25.0
	30–39	23.3 (24.5)		7.0–31.0
	40–49	28.6 (26.2)		10.0–37.0
	50–59	30.3 (26.6)		12.0–40.0
	60–69	32.1 (27.4)		13.0–42.0
	70–79	32.6 (27.0)		14.0–42.0
	80–89	32.2 (26.1)		14.0–41.0
	90–99	31.0 (24.9)		15.0–38.0
Charlson comorbidity index	0	31.1 (26.0)	<0.001	13.0–40.0
	1	29.4 (27.1)		12.0–37.0
	2	31.5 (28.2)		13.0–41.0
	≥3	36.9 (29.9)		15.0–47.0
Type of hospital	Tertiary hospital	29.4 (24.7)	<0.001	12.0–38.0
	General hospital	32.5 (27.1)		14.0–42.0
	Hospital	32.7 (27.8)		14.0–42.0
	Convalescent hospital	35.1 (35.4)		12.0–45.0
Total medical cost (KRW ×10^3^)				
Sex	Male	10,615.1 (10,199.7)	<0.001	3535.0–14,010.0
	Female	8578.6 (8883.9)		2721.0–10,949.0
Age groups (years)	20–29	6841.8 (10,172.4)	<0.001	1232.0–7159.0
	30–39	8053.2 (10,132.3)		1565.0–10,463.0
	40–49	10,139.7 (11,111.3)		2369.0–13,620.0
	50–59	10,343.9 (10,708.8)		2786.0–13,798.0
	60–69	10,691.5 (10,635.1)		3160.0–14,302.0
	70–79	9810.5 (9552.5)		3287.0–12,854.0
	80–89	8729.6 (8320.8)		3230.0–11,165.0
	90–99	7461.0 (7027.5)		2978.0–9524.0
Charlson comorbidity index	0	9525.3 (9571.3)	<0.001	3035.0–12,378.0
	1	7509.2 (8272.3)		2289.0–9447.0
	2	8632.0 (9049.7)		2684.0–11,116.0
	≥3	10,463.7 (9954.3)		3668.0–13,622.0
Type of hospital	Tertiary hospital	11,741.6 (10,959.7)	<0.001	3926.0–15,759.0
	General hospital	9142.0 (8981.5)		3063.0–11,852.0
	Hospital	5331.0 (5439.2)		3601.0–54,485.0
	Convalescent hospital	4842.1 (5223.4)		1559.0–55,191.0

CDI, *C. difficile* infection; SD, standard deviation.

**Table 5 antibiotics-13-00542-t005:** Length of hospital stay and medical cost according to comorbidity index between CDI and control group.

Length of Hospital Stay (Days)		CDI	Control	* p * Value
Comorbidity index	0	31.1 (26.0)	9.6 (10.2)	<0.001
	1	29.4 (27.1)	13.0 (12.0)	<0.001
	2	31.5 (28.2)	12.4 (12.0)	<0.001
	≥3	36.9 (29.9)	16.7 (13.7)	<0.001
Total medical cost (KRW ×10^3^)				
Comorbidity index	0	9525.3 (9571.3)	1891.2 (2193.9)	<0.001
	1	7509.2 (8272.3)	1400.7 (1430.9)	<0.001
	2	8632.0 (9049.7)	1597.6 (1892.5)	<0.001
	3	10,463.7 (9954.3)	2567.3 (2584.5)	<0.001

CDI, *C. difficile* infection; data are expressed as mean (standard deviation).

## Data Availability

Data will be available upon request from the corresponding author.

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
