# Peer review of "Demographic Characteristics and Economic Burden of *Clostridioides difficile* Infection in Korea: A Nationwide Population-Based Study after Propensity Score Matching"

_antibiotics, 2024, doi:10.3390/antibiotics13060542_

Round 1

Reviewer 1 Report

Comments and Suggestions for Authors

This is a nationwide population-based study focusing on incidence, demographic characteristics and economic burden of CDI in South Korea.

My major comment is the huge increase in CDI incidence is not investigated in more detail. Why would it have increased so much during the study period? Were infection control practices enhanced in the hospital during the study period?  More investigation of potential transmissions between patients in the hospital would be informative.

Did the number of annual bed days increase or decrease during that time period compared to average? And the annual number of CDI cases? eover, the structure of the ms should also be improved! For example, Materials and methods should be after introduction and before results and discussion.

Author Response

This is a nationwide population-based study focusing on incidence, demographic characteristics and economic burden of CDI in South Korea.

My major comment is the huge increase in CDI incidence is not investigated in more detail. Why would it have increased so much during the study period? Were infection control practices enhanced in the hospital during the study period?  More investigation of potential transmissions between patients in the hospital would be informative.

Answer: Thank you very much for your important comments. It’s a very important point. Increasing CDI incidence in our study from 2011 to 2019 is consistent with previous study, that the incidence of CDI in Asia has increased significantly from 2006 to 2014 [14]. The emergence of this epidemic may be explained by patient and virulence factors. During the study period, the mean age of patients admitted to hospital in South Korea increased considerably, as did the proportion with numerous comorbidities. In addition, as demonstrated from previous study [15], increases in CDI incidence have been largely attributed to the emergence of a previously rare and more virulent strain, which had increased toxin production and high-level resistance to fluoroquinolones. However, this study could not analyze the virulence of CDI, so additional study is needed. These contents were added in the Discussion part of revised manuscript.

  1. Ho J, Wong SH, Doddangoudar VC, Boost M, Tse G, Ip M. Regional differences in temporal incidence of Clostridium difficile infection: a systematic review and meta-analysis. Am J Infect Control 2020;48:89-94.
  2. Lessa FC, Gould CV, McDonald LC. Current status of Clostridium difficile infection epidemiology. Clin Infect Dis 2012;55:S65-S70.

Did the number of annual bed days increase or decrease during that time period compared to average? And the annual number of CDI cases? Moreover, the structure of the ms should also be improved! For example, Materials and methods should be after introduction and before results and discussion.

Answer: Thank you very much for your comments. The length of hospital stay can vary greatly depending on the cause of the disease, therefore, control group was defined as hospitalized patients who were newly diagnosed with gastroenteritis and colitis of infectious origin (A090) or unspecified origin (A099). As described in our original manuscript, the length of hospital stay was 3.3 folds greater in the CDI group than in the control group (31.5 vs 9.6 days, P < 0.001). In addition, annual number of CDI cases and CDI/10,000 person were increased during the study period.

Annual occurrence rate of CDI between 2011-2019

2011

2012

2013

2014

2015

2016

2017

2018

2019

CDI case

Total

7,438

8,257

9,268

11,079

13,112

15,648

17,000

20,283

21,762

  Male

3,391

3,731

4,183

4,887

5,905

6,901

7,577

9,207

9,958

  Female

4,047

4,526

5,085

6,192

7,207

8,747

9,423

11,076

11,804

As we know, most structure of medical journal have Materials and methods after introduction and before results and discussion. However, Antibiotics journal was unique in that Materials and methods were located after introduction, results and discussion. I modified structure of manuscript to a general format, as your comment.

Reviewer 2 Report

Comments and Suggestions for Authors

Abstract

There is an inconsistency between these two statements.

The length of hospital stay and medical costs 26 were 3.3 folds and 5.0 folds greater, respectively, in the CDI than in the control group (both P < 27 0.001). Although the length of hospital stay decreased, total medical costs increased in all age groups 28 and both sexes between 2011 and 2019 (all P < 0.001).

Introduction

In the introduction section, the author should add information about the ramifications of demographic and medical costs for CDI. Why did the author choose CDI among other causes of infectious diseases?

Methods

How did the author choose the 247.694 patients from 2.123.830 patients in the Control group?

The author should add information associated with choosing metronidazole and vancomycin history rather than clindamycin or monobactam history (Teng, 2019). Metronidazole and vancomycin are for CDI eradication.

Reference:

Teng C, Reveles KR, Obodozie-Ofoegbu OO, Frei CR. Clostridium difficile Infection Risk with Important Antibiotic Classes: An Analysis of the FDA Adverse Event Reporting System. Int J Med Sci. 2019;16(5):630-635. Published 2019 May 7. doi:10.7150/ijms.30739.

Results

What is the author's message for this sentence?

Page 3, line 90: The proportion of CDI in the 70s (31.8%) and 80s (29.4%) groups was 61.2%.

Propensity score matching is a technique to balance the confounding factor between groups. Therefore, it is expected that the proportion of the variables (confounding factor) in CDI and non-CDI is homogenous or similar after propensity matching. In Table 2, (after propensity matching) the percentage of the Charlson comorbidity index in the CDI group differs from that in the non-CDI group. Comorbidities affect the length of stay.

The author should add the length of stay and medical cost of non-CDI in Table 4 to compare it with the length of stay and medical cost of the CDI group.

Discussion

The author should discuss the big difference in CDI incidence in Korea (18.1%; 1.3%) as shown in Supplementary Figure 1. Nationwide regional distribution of CDI occurrence.

Comments on the Quality of English Language

Moderate

Author Response

There is an inconsistency between these two statements.

The length of hospital stay and medical costs were 3.3 folds and 5.0 folds greater, respectively, in the CDI than in the control group (both P < 27 0.001). Although the length of hospital stay decreased, total medical costs increased in all age groups and both sexes between 2011 and 2019 (all P < 0.001).

 Answer: Thank you very much for your comments. Sorry for confusing the two statements. The first statement is a comparison between CDI and control, and the second statement is a description for temporal changes, that is, the change between 2011 and 2019.

Introduction

In the introduction section, the author should add information about the ramifications of demographic and medical costs for CDI. Why did the author choose CDI among other causes of infectious diseases?

Answer: Thank you very much for your comments. As described in Introduction of original manuscript, CDI has been recognized as a considerable threat to public health globally, as its occurrence has increased worldwide. However, there have been insufficient data of CDI for the well-designed, PS matched analysis using nationwide data. So, we choose CDI among other causes of infectious diseases. In other Korean study reported that the medical cost of CDI was increased by 6.6 folds from 2008 to 2011. However, previous study was limited as they included indirect costs, which varies depending on each patients and accurate estimation is difficult.

   As your comments, we added information about the ramifications of medical costs for CDI as follows: Until now, only few studies have evaluated the economic burden of CDI. The economic burden of CDI is expected to increase significantly with the increasing burden of main risk factors for CDI, such as old age, antibiotic use and long-term hospital stays. South Korea is a rapidly aging country in line with global trends. In South Korea, 19.2% of the total population is elderly population older than 65 years and this proportion is expected to increase to 25.3% by 2030 [8]. Long-term hospital stays are often noticed in older patients due to comorbid diseases, frailty and malnutrition as well as CDI. In addition, a health insurance database-generated time series analysis reported the temporal change of antibiotic use from 2007 to 2014 in South Korea [8]. Defined daily doses of antibiotic use increased from 33.95 in 2007 to 37.32 in 2014 in elderly population older than 65 years and 23.5 in 2007 to 27.7 in 2014 in all age groups. As our expectation, a Korean study reported that the medical cost of CDI was increased by 6.6 folds from 2008 to 2011. However, previous study was limited as they included indirect costs, which varies depending on each patients and is difficult to accurately estimate. Considering the increasing incidence of CDI and associated economic burden, updated nationwide, economic burden of CDI should be evaluated.

[8] Statistics Korea. 2024 Elderly statistics. Statistics Korea. 2024. http://kostat.go.kr/portal/korea/kor_nw/1/1/index.board?bmode=read&aSeq=385322 Accessed 29 May 2024.

[9] Park J, Han E, Lee SO, Kim HS. Antibiotic use in South Korea from 2007 to 2014: a health insurance database-generated time series analysis. PLoS One 2017;12:e0177435.

Methods

How did the author choose the 247.694 patients from 2.123.830 patients in the Control group?

Answer: Thank you very much for your comments. It was described in the original manuscript. Among 2,123,830 control group who met the eligible criteria before PS matching, and 247,694 control patients were included after 1:2 PS matching with 123,847 CDI cases after matching for sex, age, age groups and year of hospitalization. In addition, ‘MatchIt’ package of the R program was used for PS matching between two groups.

The author should add information associated with choosing metronidazole and vancomycin history rather than clindamycin or monobactam history (Teng, 2019). Metronidazole and vancomycin are for CDI eradication.

Reference:

Teng C, Reveles KR, Obodozie-Ofoegbu OO, Frei CR. Clostridium difficile Infection Risk with Important Antibiotic Classes: An Analysis of the FDA Adverse Event Reporting System. Int J Med Sci. 2019;16(5):630-635. Published 2019 May 7. doi:10.7150/ijms.30739.

Answer: Thank you very much for your comments. Teng C, et al. evaluated the association between CDI and important antibiotic exposure, such as clindamycin or monobactam. They analyzed the risk factors of CDI. However, out study analyzed the confirmed CDI cases, who was diagnosed and treated as CDI rather than risk factor, so it was defined as a combination of CDI diagnosis coding (A047) and treatment medications (metronidazole and vancomycin, which are eradication regimen of CDI).

Results

What is the author's message for this sentence?

Page 3, line 90: The proportion of CDI in the 70s (31.8%) and 80s (29.4%) groups was 61.2%.

Answer: Thank you very much for your comments. I’m simply describing that the incidence of CDI in elderly patients over 70s is very high as 61.2% (31.8% in 70s and 29.4% in 80s) of all CDI incidences.

Propensity score matching is a technique to balance the confounding factor between groups. Therefore, it is expected that the proportion of the variables (confounding factor) in CDI and non-CDI is homogenous or similar after propensity matching. In Table 2, (after propensity matching) the percentage of the Charlson comorbidity index in the CDI group differs from that in the non-CDI group. Comorbidities affect the length of stay.

The author should add the length of stay and medical cost of non-CDI in Table 4 to compare it with the length of stay and medical cost of the CDI group.

Answer: Thank you very much for your critical comment. I agree with your opinion. In our study, we used ‘MatchIt’ package in the R program for PS matching for sex, age, age groups and year of hospitalization between two groups. In our study, Charlson comorbidity index was not included as a matching variable because Charlson comorbidity index was very different two groups, and two groups were not well balanced even after PS matching when Charlson comorbidity index included. As shown in Table 4, comorbidities affect the length of stay in CDI. Length of hospital stay of CDI was significantly increased in comorbid patients (36.9 days in ≥ 3 comorbidity index vs 31.1 days in comorbidity index 0, p < 0.001).

Table 4 shows the length of stay and medical cost in patients with CDI, according to sex, age groups, comorbidity index, and type of hospital. So, it is difficult to present e comparison of length of stay and medical cost between CDI group and control group within Table 4. So, we added supplementary Table 1 to show the effect of comorbidity in both groups. When comparing the length of hospital stay and medical cost according to comorbidity index between CDI and control group (Suppl. Table 1) as comorbidities affect the length of stay and medical cost, they were significantly greater in CDI than control group in all comorbidity scores (all p < 0.001). The length of hospital stay was longer by 2.2-3.0 folds and medical cost was also higher by 4.1-5.4 folds in CDI than control group regardless of comorbidity scores. However, the length of hospital stay was not significantly affected by comorbidity scores. They are additionally described in Results.

Supplementary Table 1. Length of hospital stay and medical cost according to comorbidity index between CDI and control group.

Length of hospital stay (days)

CDI

Control

P value

Comorbidity index

0

31.1 (26.0)

9.6 (10.2)

< 0.001

1

29.4 (27.1)

13.0 (12.0)

< 0.001

2

31.5 (28.2)

12.4 (12.0)

< 0.001

≥ 3

36.9 (29.9)

16.7 (13.7)

< 0.001

Total medical cost (x103 won)

Comorbidity index

0

9,525.3 (9,571.3)

1,891.2 (2,193.9)

< 0.001

1

7,509.2 (8,272.3)

1,400.7 (1,430.9)

< 0.001

2

8,632.0 (9,049.7)

1,597.6 (1,892.5)

< 0.001

3

10,463.7 (9,954.3)

2,567.3 (2,584.5)

< 0.001

Data are expressed as mean (SD).

Discussion

The author should discuss the big difference in CDI incidence in Korea (18.1%; 1.3%) as shown in Supplementary Figure 1. Nationwide regional distribution of CDI occurrence.

Answer: Thank you very much for your important comment. As your comment, we noticed a big difference in the CDI incidence within Korea. Because a lot of content was covered in the original draft, this issue was not described in detail. The regional difference in CDI occurrence appears to be due to regional differences in hospital distribution rather than actual difference in incidence rate. In our PS matched analysis in Table 2, 34% and 52.3% of CDI cases were detected in tertiary and general hospitals, respectively. In Korea, there is an unbalanced regional distribution of tertiary and general hospitals: 29.8% of tertiary hospitals and 13.1% of general hospitals are located in Seoul, where 18.1% CDI was reported. Whereas, 2.1% of tertiary hospitals and 2.5% of general hospitals are located in Ulsan city, where only 1.3% CDI was reported. It was described in the Discussion.

Round 2

Reviewer 1 Report

Comments and Suggestions for Authors

The authors  provided a point-by-point response in all comments and modified the structure of the ms.